# Heart Rehabilitation for All (HeRTA): Protocol for a feasibility study and pilot randomized trial

Hanne Birke[1,☯]*, Ida Foxvig[1☯], Karin Burns[1☯], Ulla Toft[1,2☯], Anders Blædel Gottlieb Hansen[1☯], Pernille Ibsen Hauge[3‡], Sussie Foghmar[4‡], Rikke Bülow Mindegaard[5‡], Louise Meinertz Jakobsen[1,6☯]

1 Center for Clinical Research and Prevention, Frederiksberg Hospital, Frederiksberg, Denmark, 2 Section of Social Medicine, Department of Public Health, University of Copenhagen, Copenhagen, Denmark, 3 The Danish Heart Association, Copenhagen, Denmark, 4 Cardiology Department, University Hospital Hvidovre, Hvidovre, Denmark, 5 Rehabilitation Center, Albertslund Municipality, Albertslund, Denmark, 6 Capital Region, Hillerød, Denmark

☯ These authors contributed equally to this work.
‡ PIH, SF and RBM also contributed equally to this work.
* Hanne.birke.01@regionh.dk

## Abstract

### Introduction

Today, 50% of people with cardiac disease do not participate in rehabilitation. The HeRTA-study aims to develop and test a sustainable rehabilitation model supporting vulnerable patients in participating in rehabilitation and long-term physical activity.

### Methods

A feasibility study with a non-blinded pilot randomized trial was developed in collaboration with partners and cardiac patients to test a multi-component rehabilitation intervention across hospital, municipality, and civil society. The study runs from January 2020 to December 2024. Eligibility criteria for participants: a) diagnosed with either ischemic heart disease, persistent atrial fibrillation, heart failure, or have had cardiac valve surgery, b) residents in Hvidovre Hospitals uptake area, c) cognitively functional, d) physically able to participate in rehabilitation. Patient recruitment will be located at Hvidovre Hospital, Capital Region of Denmark, data collection at Hvidovre Hospital, Rehabilitation Center Albertslund, the Danish Heart Association, and in two municipalities (Hvidovre and Brøndby). Patients in the control group have access to usual care at the hospital: rehabilitation-needs-assessment, patient education, and physical training. After or instead of hospital rehabilitation, the patient can be referred to municipal rehabilitation with patient education, and a total of 12 weeks of physical training across sectors. Patients in the intervention group will in addition to usual care, have access to *an information book* about cardiac disease, *patient supporters* from the Danish Heart Association, Information materials to inform employers about the employees' rehabilitation participation, a *rehabilitation goal setting plan*, a *support café for relatives*, *and follow-up phone calls* from physiotherapists 1 and 3 months after rehabilitation to

registrations. The participants in this study have not given consent to make these data publicly available. A request for access to the data needs approval from appropriate Danish authorities and is subject to Danish regulations on personal data protection. A request for an arrangement of data transfer agreements can be sent to the corresponding author Hanne Birke or to Michaela Louise Schiøtz, e-mail: michaela.louise. schioetz@regionh.dk.

**Funding:** This work was supported by The Danish Heart Association (19-R136-A9125-22129 to HB) https://www.hjerteforeningen.dk, by Brødrene Hartmann (A36031 to HB) https://www. hartmannfonden.dk/, and by Helsefonden (21-B-0016 to HB) https://www.helsefonden.dk. The funders had no role in study design, data collection and analysis, decision to publish, or preparation of the manuscript.

**Competing interests:** The authors have declared that no competing interests exist.

support physical activities. Patients with vulnerabilities will additionally receive *patient education conducted in small groups*, *pro-active counselling* by a cardiac nurse, psychologist, or social worker, *paid transportation* to rehabilitation, and *paid membership* in a sports association. Patients are computer block-randomized so patients with vulnerability are distributed evenly in the two study arms by stratifying on a) a cut-off score of $\geq 5$ in the Tilburg Frailty Indicator questionnaire and/or b) need of language translator support. A power calculation, based on an estimated 20% difference in participation proportion between groups, 80% power, a type 1 error of 5% (two-tailed), results in 91 participants in each study arm. The primary outcome: rehabilitation participation (attending $\geq$ two activities: patient education, smoking cessation, dietary counseling, and physical training) and reaching at least 50% attendance. Secondary outcomes: health-related quality of life, coping strategies, level of physical activities, and sustainability regarding participation in active communities after rehabilitation. The study is registered at ClinicalTrials.gov (NCT05104658).

## Results

Differences between changes in outcomes between groups will be analyzed according to the intention-to-treat principle. Sensitivity analysis and analysis of the effect of the combined activities will be made. A process evaluation will clarify the implementation of the model, the partnership, and patients' experiences.

## Conclusion

Cross-sectoral collaborations between hospitals, municipalities, and organizations in civil society may lead to sustainable and affordable long-term physical activities for persons with chronic illness. The results can lead to improve cross-sectoral collaborations in other locations and patient groups.

## Introduction

Cardiovascular diseases are the leading cause of death globally accounting for 17.9 million deaths in 2019 amounting to 32% of all global deaths [1]. In Europe, cardiovascular diseases cause more than half of all deaths [2]. In Denmark, half a million people, corresponding to one out of ten, suffer from cardiovascular diseases [3] which often coincides with other chronic conditions [4].

Cardiac rehabilitation is essential to diminish the consequences of cardiovascular diseases and prevent new cardiac episodes [5]. Solid evidence exists on the benefits of multifaceted cardiac rehabilitation on patients' cardiovascular function [5], physical functional level [6, 7], quality of life [8, 9], and survival [8, 9]. The Danish rehabilitation program is inspired by the chronic care model and is initiated at the hospital and finalized in at a rehabilitation center in a municipality [10]. The Danish cardiac rehabilitation program includes systematic efforts to improve health quality, patient involvement, and patient satisfaction by offering disease management, medical treatment, tobacco cessation support, physical exercise, psychosocial support, and guidance about alcohol and nutrition [11].

Today, only half of Danish cardiac patients participate in rehabilitation, and vulnerable patients are more likely to decline participation compared to more resourceful patients [12].

Structural barriers during the transition between hospital and municipality make it difficult for these patients to navigate in rehabilitation activities as the activities are in different locations and different sectors. A Danish cohort study found that 39% of patients commencing rehabilitation activities dropped out during rehabilitation–and 70% of these dropouts happened at the transition phase from hospital to municipality [13]. Other barriers to participation are a poor financial situation, weak social relationships, logistic challenges, language difficulties, and cultural considerations [12, 14–16].

The probability of being offered rehabilitation is lower if patients live alone, are unemployed, have a short education, have a low income, or suffer from several chronic conditions [12]. Despite health professionals' intentions to address vulnerability, vulnerable patients are overrepresented among those who do not receive a referral to rehabilitation [17], participate in rehabilitation [18] or complete rehabilitation activities [18]. Even after having participated in rehabilitation programs, many patients struggle to maintain new lifestyle habits [17] which will be in focus in present study.

## Overall objectives

The overall aim of Heart Rehabilitation for All (HeRTA) is to develop and test the feasibility of a new, sustainable model for rehabilitation supporting vulnerable patients to take part in rehabilitation and promoting long-term activity for all patients with cardiac disease.

More specifically our goal is to:

- develop and implement differentiated rehabilitation interventions

- test if combined activities across sectors can increase the proportion of cardiac patients' participating in rehabilitation

- test if the model improves the maintenance of lifestyle changes and enhances physical and mental functioning, quality of life, and self-care capacity among cardiac patients

- test whether formation of a partnership has a positive influence on development and implementation of the HeRTA model

- test how organizational context, possible changes thereto, and individual approaches among professionals influence the implementation of the model

- test the feasibility of this model for rehabilitation

## Conceptual framework

We are inspired by the complex intervention framework developed by the UK Medical Research Council (MRC) [19], as cardiac rehabilitation research demands cross-sectoral collaboration and action involving health care professionals in several health care organizations. To connect health care professionals, a non-governmental organization, and an advisory board of patients with cardiac disease, we enrolled them in a partnership. The overall partnership theory is that by working together, partners can achieve more than they can on their own [20, 21]. Partnerships as a means to create collaboration for the benefit of health were conceptualized in 1978 [22] and intersectoral partnerships have grown to be an integral part of health promotion research, practice, and policy [20, 21].

Using a framework of complex intervention research is useful when examining research questions about rehabilitation course [23].

Several guidelines on complex intervention have been published. In 2006, the MRC published a revised framework on developing and evaluating complex interventions [19]. Later in

2021, the National Institute of Health Research (NIHR) and the MRC published a further developed framework comprising conceptual, methodological, and theoretical developments [23]. Complex interventions are delivered and evaluated at different stages, from individual to organizational levels [23]. To consider the complexity in complex interventions rising from the components in the intervention and the contextual interaction during implementation provide beneficial knowledge for decision makers [23].

To strengthen involvement and ownership by all partners, collaborators, and patient representatives, we are inspired by frameworks for co-creation and intervention development by Hawkins et al. [24], Meroni et al. [25], and O'Cathain et al. [26]. The INVOLVE guideline [27] has guided us on ensuring equality in the process. Co-creation is a method of creating a joint approach to rehabilitation involving activities from both the public health sector and civil society. In addition, co-creation supports local ownership of the developed activities.

The project applies a system thinking approach, that encourages people to focus their attention on how different 'agents' (people, services, organizations, etc.) interconnect and influence each other [28]. Throughout the co-creation, implementation, and evaluation of the new multi-component intervention, we acknowledge the project's interference on normal rehabilitation pathways and are alert to new developments that may require changes to the evaluation during the project [28].

## Methods

### Design

The study is a feasibility study with a pilot randomized trial. The design is chosen to test the feasibility of the rehabilitation model, including cross-sectoral collaboration and implementation before upscaling to a larger randomized controlled trial.

The project is organized in three phases:

1. **A development phase** (1st quarter 2020 to the 1st quarter 2022), where all partners including the patient advisory board has participated in a partnership and co-creation process that resulted in development of model content and collaboration procedures.

2. **A feasibility phase** (2nd quarter 2022 to the 3rd quarter 2023) with a small scale randomized controlled trial (RCT)-component (Fig 1), where we will examine whether the intervention activities are feasible, acceptable and may have positive effects for patients with heart disease. Furthermore, we want to assess the feasibility of the future evaluation design by exploring recruitment methods, data collection, participant retention strategies, the willingness of participants to be randomized, randomization procedures, risk of contamination, and preliminary cost calculation of the intervention.

3. **A two-year-follow-up phase** (2nd quarter 2023 to the 4th quarter 2024), in which sustainability of the intervention on patient activity will be assessed and promising components will be further tested.

### Ethics

A data handling plan has been accepted by the Knowledge Center for Data Reviews, the Capital Region of Denmark (journal-nr.: P-2020-905, date 12-11-2021). Approval is not needed from the Danish National Committee of Health Research Ethics [30]. The study is registered at ClinicalTrials.gov [31], identifier: NCT05104658.

We will conduct the study in line with the ethical principles for medical research as described in the Declaration of Helsinki [32]. During the testing phase, the security of the

| | STUDY PERIOD | | | | | | | |
|---|---|---|---|---|---|---|---|---|
| | Enrolment | Allocation | | Post-allocation | | | Close-out | |
| Time points | One week after discharge | Base line | 3 days before RNA* | RNA* | 3 months follow-up | 6 months follow-up | 12 months follow-up | 24 months follow-up |
| Eligibility screen | X | | | | | | | |
| Informed consent | X | X | | | | | | |
| Reminder or help to fill out baseline | | | X | | | | | |
| INTERVENTIONS: | | | | | | | | |
| Usual care | | | | ←——→ | | | | |
| Multi-component rehabilitation model | | | | ←————→ | | | | |
| ASSESSMENTS: | | | | | | | | |
| Physical, mental, and social vulnerability | | X | | | | | | |
| participation in rehabilitation activities | | | | X | X | | | |
| Positive and active engagement in life | | X | | | X | X | | |
| Physical and mental health summaries | | X | | | X | X | X | X |
| Physical activities in leisure time | | X | | | X | X | X | X |

**Fig 1. Time schedule of enrolment, interventions, and assessments on participant outcome inspired by the SPIRIT 2013 reporting guidelines [29].** *Rehabilitation needs assessment.

anonymity of respondents is not possible due to the close collaboration between the health care professionals and the patients to provide individualized rehabilitation offers. However, quantitative results will be presented in accumulated form and care will be taken to guarantee that no respondents are recognizable in the results. All data from questionnaires and registration of rehabilitation participation will be securely stored and deleted after completion of the study.

## Project organization

To create close and reciprocal bonds between the involved agents from different sectors HeRTA used a partnership approach [33] and was organized as a partnership between the Cardiac Outpatient Clinic at Hvidovre Hospital, the rehabilitation center in Albertslund municipality, the Danish Heart Association, The Intersectoral Prevention Laboratory, and the Center for Clinical Research and Prevention (CCRP). Other stakeholders participate at a collaborative and counseling level (see Fig 2 and list below).

The role of the participants:

- The CCRP initiated the project, decided, and managed the chosen research methods and content, facilitated the development processes, and will conduct the future scientific analyses and evaluations

- The Intersectoral Prevention Laboratory participated in the initiation of the project and provided guidance to the chosen research methods

- The Cardiac Outpatient Clinic at Hvidovre Hospital participated in the development phase, and will take part in the recruitment of patients and in delivering the developed rehabilitation services

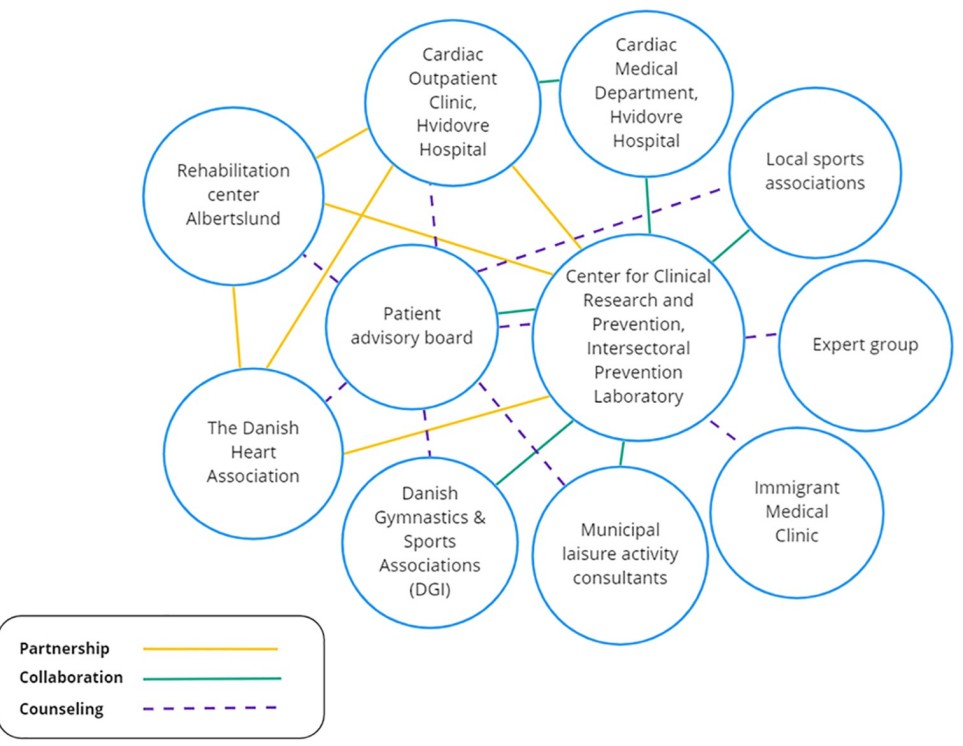

**Fig 2. Project organization.**

- The rehabilitation center in Albertslund municipality participated in the development phase and will deliver the developed rehabilitation services

- The Danish Heart Association participated in the development phase and will deliver the developed rehabilitation services

- The patient advisory board consisting of 11 citizens with heart disease, is involved at a counseling level throughout the project to ensure patient involvement and a continued focus on patient needs. They participated in the development phase and will be consulted throughout the remainder of the project

- Consultants from Brøndby and Hvidovre municipalities, the Danish Gymnastics & Sports Associations and volunteers from local sports associations participated at a collaborative level in the development of bridge-building efforts in the transition from municipality to physical activities in local communities. Local sports associations will provide various activities to citizens after completing rehabilitation

- The expert group and the Immigrant Medical Clinic provides counseling on the choice of research methods, outcome measurements, collection and processing of data, analyses, and involvement of non-Danish speaking patients. The members have expertise in complex real-life health interventions, cardiac rehabilitation, cross-sectional knowledge transmission, patient involvements, partnerships and co-creation, and migrant health and rehabilitation.

HeRTA will assess the course of the partnership formation, the influence of the partnership on the development of rehabilitation activities, and the subsequent implementation. During the implementation, patients' benefits of the partnership in terms of coherent patient pathways and coordinated activities will be assessed. Monitoring and evaluation of the partnership will

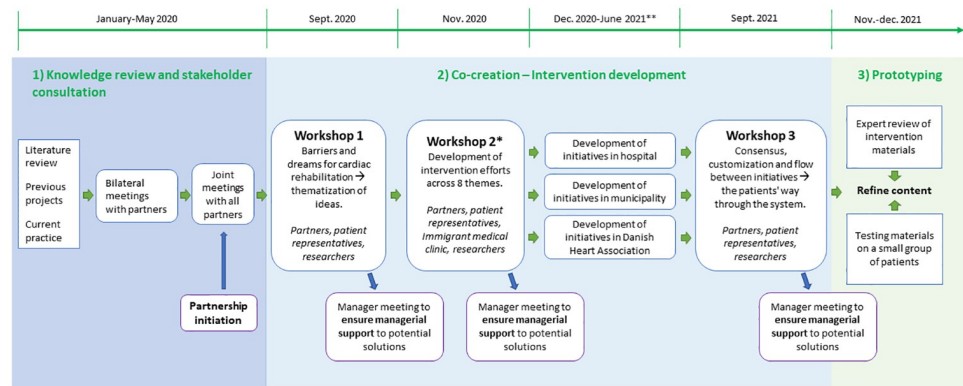

**Fig 3. Framework for co-creation and prototyping in a complex intervention.** *Virtual workshop do to Covid-19.
**The Cardiac Outpatient Clinic staff were moved to Covid-care (January-April).

be made by the internationally validated Partnerships Analysis Tool, developed by The Victorian Health Promotion Foundation.

(VicHealth) in Australia [33]. The tool was translated and adapted to a Danish context in 2019 [34]. The tool aims to help the partnering process forward by making the involved organizations reflect on the partnership and monitor its effectiveness. HeRTA will test the practical relevance of using the instrument to support and develop a partnership in a Danish rehabilitation context.

## Development of the complex intervention

The partnership and co-creation processes were conducted (1st quarter 2020 to the 1st quarter 2022) and included the following steps (Fig 3):

1. **Stakeholder meetings.** The aim of this step was knowledge building and partnership initiation. At the joint meetings, we learned about the partners' terms of collaboration as well as getting a common understanding of the bigger picture (the system).

2. **Co-creation–intervention development** with all partners and patient representatives. **Workshops:**

*In workshop 1*, insights from the dialogue meetings in step 1 were presented. Partners and patients were actively involved through a facilitative development process based on design thinking [35]. The product from workshop 1 was ideas and visions for intervention activities and incipient thematization.

*In workshop 2*, partners, and patients brought together again in a co-creative process. This workshop was online due to Covid19. Participants worked in mixed groups on 8 specific themes from workshop 1, that got the most votes out of 14. At this point, we also involved the immigrant medical clinic due to their knowledge of handling linguistic/cultural barriers and interpretation among non-Danish speaking patients. The product of workshop 2 was a rough sketch of the intervention efforts. Afterward, CCRP conducted interviews with vulnerable cardiac patients to assess the match between intervention components and patient needs.

*In workshop 3*, each partner presented their intervention efforts and got feedback from the partners and patient representatives to ensure coherency. The product of workshop 3 was finalized recruitment strategies, referral procedures, and rehabilitation content.

**Development meetings:** As the complex rehabilitation intervention consists of efforts made by various partners, time was set aside between workshops 2 and 3 for the partners to further develop their efforts separately based on results from workshop 2. Patient representatives and researchers were invited to give feedback and facilitate.

**Managerial support:** After each workshop, a meeting was set up with the managers from each partner. At these meetings, workshop results were presented, reviewed, and decisions were made about which activities to proceed with.

3. **Prototyping.** The rehabilitation content was described in an intervention manual (prototype) and assessed by HeRTAs expert group on acceptability and feasibility. The completion of Tilburg Frailty Indicator questionnaire was tested on 20 cardiac patients to assess the prevalence of those meeting a cut-off score of $\geq 5$, which is recommended as an indicator of vulnerability [36, 37]. The final decision to initiate the feasibility study with a pilot randomized trial was taken by management in all organizations.

## The pilot randomized trial

**Setting.** The study takes place in the uptake area of Hvidovre University Hospital in the Capital Region of Denmark [38] and recruitment runs from 2nd quarter 2022 to the 3rd quarter 2023. The hospital receives approximately 300 cardiac patients a year, out of which around 200 patients are provided cardiac rehabilitation at Rehabilitation Center Albertslund. Patient recruitment will be located at Hvidovre University Hospital, data collection will be located at Hvidovre University Hospital, Rehabilitation Center Albertslund, the Danish Heart Association, and in active communities in two municipalities (Hvidovre and Brøndby).

**Power calculation.** A power calculation, based on an estimated 20% difference in participation proportion between groups, 80% power, and a type 1 error of 5% (two-tailed), showed that 91 participants in each group must complete the study. As we expect a 20% dropout (23 participants), 114 participants must be included in intervention and control group. If needed, the inclusion period will be extended to reach the calculated numbers of participants. Dropouts are defined as deaths, readmissions, change of address away from Brøndby or Hvidovre municipality, withdrawal of consent, and patients' discontinuing questionnaire completion during the intervention period. See flow chart in Fig 4.

**Inclusion procedure.** Eligible patients are:

a) Diagnosed with either ischaemic heart disease, persistent atrial fibrillation, heart failure, or have had cardiac valve surgery

b) Residents in Hvidovre Hospitals uptake area

c) Cognitively functional

d) Physically able to participate in rehabilitation activities

All eligible patients will be identified by a clinical coordinator in collaboration with nursing staff. Patients will be contacted in person during admission or by phone within 1 week after discharge. Patients accepting the invitation to participate will receive a link to the project database in the Research Electronic Data Capture (REDCap) [39], a web-based application developed to capture data for clinical research. In REDCap they are asked to consent to a) information exchange between partners and b) participation in the project. We follow the Danish Data Protection Agency requirements to consent forms and obligation to inform participants [40]. Furthermore, they will be asked to complete a baseline questionnaire. If patients

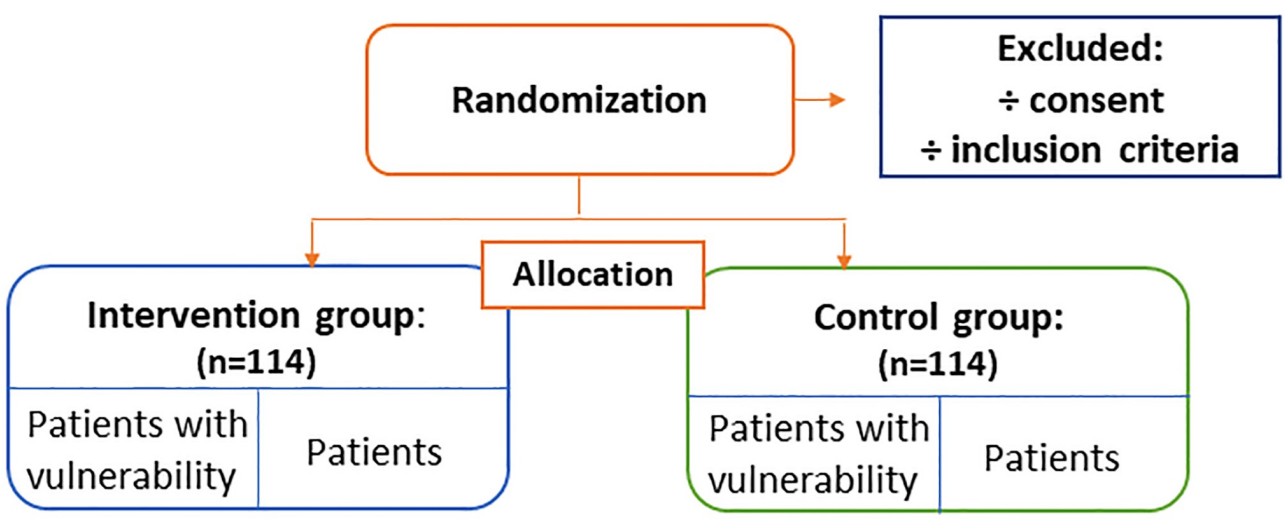

**Fig 4. Flow chart.**

have not completed the questionnaire 3 days before their rehabilitation needs assessment consultation with a nurse, they will receive a reminder on phone and if necessary, assistance to fill out the baseline questionnaire. At 3, 6, 12, and 24 months after baseline, the patients will receive up to two SMS reminders to promote completion of the questionnaires. Vulnerability is assessed at baseline and defined as either a) a cut-off score of $\geq 5$ [36] in the Tilburg Frailty Indicator questionnaire [37] or b) need of language translator support during consultations.

**Randomization procedure.** Patients are block-randomized automatically in REDCap [39], so patients with vulnerability are distributed evenly in the control and intervention groups by stratifying on a) a cut-off score of $\geq 5$ [36] in the Tilburg Frailty Indicator questionnaire [37] and b) need of language translator support during consultations. The clinical coordinator ensures that patients are allocated accordingly to the group of nurses providing usual care or intervention activities.

**Intervention.** For both groups, the rehabilitation process is initiated approximately 14 days after discharge at a rehabilitation needs assessment consultation with a trained cardiac nurse at Hvidovre University Hospital. The two groups are offered usual rehabilitation care to avoid impairing the usual rehabilitation care and since is inappropriate not to offer the current rehabilitation care.

**Patients in the control group** have access to usual rehabilitation course at the cardiac outpatient clinic: rehabilitation needs assessment with guidance and medical information, lecture on dietary needs (2x3 hours), a cardiac education (2x3 hours), and physical activity. After or instead of hospital rehabilitation, the patient can be referred to municipal rehabilitation in Rehabilitation Center Albertslund which offers patient education for 3x2 hours by a cardiac nurse and 1x2 hours by a dietician. Patients have access to training (1 hour 2 times a week) for a total of 12 weeks across sectors. Depending on the severity of the condition the training is either fully in the hospital, divided evenly between hospital and municipality, or fully in the municipality.

**Patients in the intervention group** will, in addition to usual care, have access to the following activities (for an overview see Fig 5):

At the rehabilitation needs assessment, all patients will receive an *information book* that provides contact information to all partners as well as relevant and thorough information on

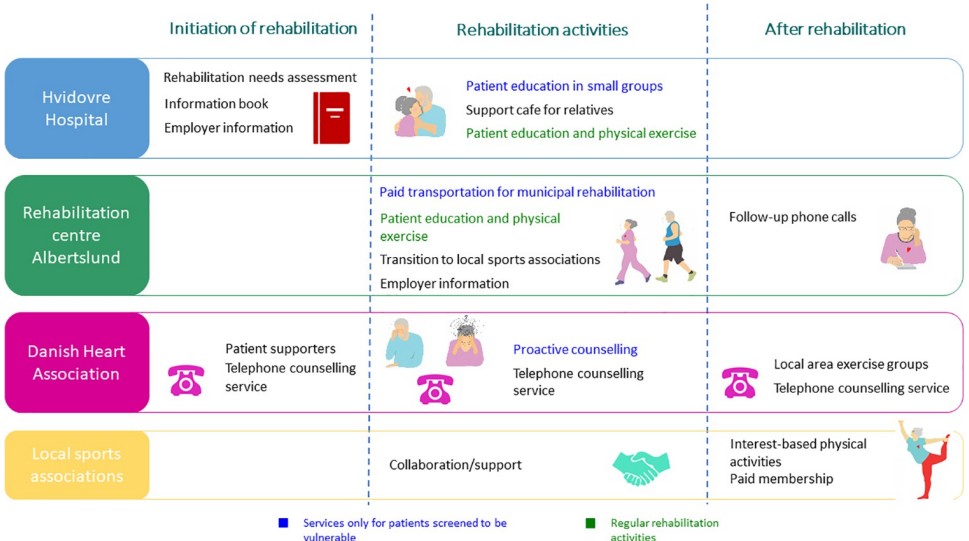

**Fig 5. Overview of the content of the intervention.**

cardiac disease, public rehabilitation, and physical training in the local community. The information book also contains information on the availability of _patient supporters_ from the Heart Association that provide a 1:1 conversation to patients who wish to talk to an experienced patient with heart disease about their worries concerning life with cardiac disease. _Employer material_ will help patients inform employers and gain their support for rehabilitation participation and a plan for returning to work. During the rehabilitation needs assessment the cardiac nurse helps the patient set _clear and reachable goals_ for rehabilitation. Relatives are informed about the _support café for relatives_ that aim to provide support in dealing with their worries as well as in supporting cardiac patients.

Patients with vulnerabilities will also have access to _patient education conducted in small groups_ that utilize patient involving methods to target and provide information relevant to patients' daily lives. In addition, patients with vulnerabilities may be referred to _pro-active counselling_ where a cardiac nurse, psychologist, or social worker from the Heart Association calls the patient to provide targeted support to their specific worries and challenges with the purpose of motivating the patient to participate in the rehabilitation program.

Patients with vulnerabilities may receive _paid transportation_ to ensure that finances are not an obstacle for commuting to Rehabilitation Center Albertslund. Finally, they will be reimbursed for the cost of 6 months membership in a sports association.

To support the transition between sectors, information on each patient is noted in "Your rehabilitation plan" and based on patient consent communicated to the municipality, and if relevant The Danish Heart Association.

During rehabilitation at the hospital and municipality, collaboration with civil society provides an easy _transition to local sports associations_ and the Heart Association's _local area exercise groups_. _Follow-up phone calls_ from the patient's municipal contact person 1 and 3 months after rehabilitation support patients to continue physical activity.

## Data collection

### Patient outcomes

The study's **primary outcome** is rehabilitation participation defined as:

✓ attending ≥ two activities (patient education, smoking cessation, dietary counseling, and physical training) and reaching at least 50% turn up.

**Secondary outcomes** are health-related quality of life, coping strategies, level of physical activities, and sustainability in terms of participation in local sports associations or exercise groups after rehabilitation.

Following data will be collected for the non-participators: age, sex, and whether they are cognitive functionable.

## Registration of participation and patient outcomes

Partners register patients' participation in all activities to monitor participation rates across sectors. Data from self-reported patient questionnaires are collected at baseline (background, and outcomes), 3, 6, 12, and 24 months (outcomes, and participation in rehabilitation activities). Physical activity in local community participation are also collected at 12 and 24 months after baseline.

**Questionnaires include:**

- Tilburg Frailty Indicator [37]: physical, mental, and social vulnerability.
  Health Education Impact Questionnaire (HEIQ) [41]: positive and active engagement in life.

- 12-Item Short Form Survey (SF-12) [42]: Physical and mental health summaries (PCS and MCS).

- Nordic Physical Activity Questionnaire-short (NPAQ) [43]: physical activities in leisure time.

## Qualitative data on patient experiences

The research team conducts qualitative semi-structured interviews with 20 patients to uncover their experience of the overall process, contact with health care professionals, the match between activities and rehabilitation needs, patients' satisfaction with heart disease education, and intersectoral coordination. Patients will be selected based on criteria for maximum variation. Both patients with high and low levels of participation will be selected for interviews.

## Qualitative data on implementation among professionals

Organizational characteristics and cultures as well as individual approaches among professionals will shape the implementation of activities. Field notes on adjustments in the intervention content, procedures, and changes in the context (e.g. organizational changes, changes in management/employees) will provide knowledge on the setting and the processes affecting the potential for the effect of the rehabilitation model. Observations and/or recordings of rehabilitation activities in Hvidovre hospital, Albertslund municipality, and the Danish Heart Associations' proactive counseling will provide insight into the actual content of the activities and the fidelity of the implementation. Focus groups with involved partners will uncover the professional's experiences and reflections on screening and referral procedures, information exchange, intersectoral collaboration, and rehabilitation activities.

## Monitoring

As the content of the intervention does not have a risk of adverse effect on the participants, a data monitoring committee is not needed nor is a plan for stopping the intervention in case of adverse events.

**Adjustments during the implementation.** To ensure successful implementation and sustainability, partners will meet quarterly to evaluate experiences with patient pathways and collaboration across partners. At these meetings, necessary adjustments are agreed upon. In the initial three months, meetings are more frequent to allow for relevant adjustments. In the remaining 9 months, a strong rationale for changes should be provided to protect the potential of the study's RCT component. All changes in activities or procedures throughout the feasibility phase—including the rationale for adjustments—will be registered in field notes.

## Data management

The qualitative data will be secured in closed folders in the Capital Region of Copenhagen's server to which only members of the research group in CCRP have access. Only the principal investigator (HB) has access to the folder containing the conversion key. The quantitative data is secured in the project database in the REDCap [38]. A data management agreement and a collaboration contract has been mutually accepted by all involved partners. The CCRP is the only partner with access to the final trial dataset.

## Analyses

**Analyses of the effect sizes.** We will perform descriptive analyses of baseline characteristics. Descriptive data will be analyzed as means with standard deviations, medians with interquartile ranges, or frequencies with percentages, depending on the distribution of variables. Differences in outcomes between the intervention and control groups will be analyzed. Multiple imputation will be used to handle missing data in case of dropouts where patients discontinue questionnaires completion during the intervention period´. Imputation will be based on age, sex, diagnose, vulnerability, and participation in rehabilitation activities. Dropouts because of deaths, new admissions, withdrawal of consent or change of address away from Brøndby or Hvidovre municipality will be excluded from the analyses. Furthermore, sensitivity analyses will be carried out. If the power of data allows—explorative subgroup analyses will be carried out on those screened to be vulnerable. Estimated effect sizes will be calculated to inform future assessment of sample sizes in RCTs.

The analyses will test the effect of the combined activities and consequently not allow identification of any decisive "ingredient". Analysis of the different combinations of activities will shed light on the activities highly or less likely used.

**Qualitative evaluations.** The project is inspired by O'Cathain et al. 's guidance for using qualitative research in feasibility studies for trials [44]. The guidance contains recommendations on how to choose the best research questions, consideration of using a range of appropriate qualitative methods and approaches addressing key feasibility questions, attention to participant sample diversity, the timing of analysis in stages in a dynamic approach, focus on a few key issues in the analysis and describing the qualitative analysis and findings.

All data from interviews and focus groups will be audio-recorded and transcribed verbatim. Interview transcriptions and field notes will be analyzed using systematic text condensation described by Malterud [45]. The analytical process will be inspired by principles from Collaborative Data Analysis [46].

**Patients' experiences from the intervention.** The analysis will assess whether patients experience their treatment and rehabilitation course as an integrated and coordinated effort helping them to live with their heart disease. The analysis will nuance the quantitative analysis and assess whether the model succeeds in tailoring rehabilitation activities to individual needs [44]. The full rehabilitation package is not necessary or relevant for all patients with heart disease. Patients with vulnerabilities may need elaborate support and encouragement to deal with

the mental and physical challenges of heart disease, while patients with resources may need less assistance. Rather they may need to be supported in continuing making healthy lifestyle choices and in using their local area and network to return to everyday life. In this analysis, gaps in the match between activities and patient needs will be identified.

**The implementation processes.** The analyses of organizational experiences are inspired by the context, process, and outcome evaluation model for organizational health interventions [47]. The analyses will focus on the fit between the varying partnering institutions and the developed rehabilitation activities and collaboration procedures. The acceptability of the intervention among employees will be assessed to ensure their support in the implementation. Organizational changes in the partnering institutions that can affect the implementation will also be assessed. Adjustments, reach, and fidelity of the implementation will be assessed to provide relevant knowledge for interpreting and drawing conclusions on the potential of each intervention component.

**Cost-effectiveness analyses.** To compare the costs of the intervention towards the effects, a preliminary cost-effectiveness analysis will be carried out. This will be done by calculating a ratio where the denominator is the health gains measured by quality-adjusted years of life (SF-12/SF-6D) and the numerator is the cost associated with the health gain obtained from the intervention, which will be the total cost of the intervention including health care resources used by the patients [48–50]. Register data from Statistics Denmark [51] will be used to calculate the cost of health care resources.

## Discussion

Improved cardiac rehabilitation actions supporting sector transitions and participation are central in reducing the consequences of cardiovascular diseases [5–9]. In most studies investigating cardiac rehabilitation, the group most likely to decline participation are persons with vulnerabilities such as multimorbidity, co-morbidity, cultural barriers, and low socioeconomic status. As cardiac rehabilitation is cross-transitional involving different health care sectors, health care professionals, and organizations, a partnership approach can be useful to create bonds between the participants. Furthermore, in the development of new rehabilitation services, co-creation as a method of creating a joint cross-sectoral approach to rehabilitation supports local ownership [24–26]. The present rehabilitation model can contribute to remedying social inequality in rehabilitation by supporting participation among cardiac patients with vulnerabilities.

### Limitations

The study is a feasibility study and although it has an RCT component is not focused on estimating effect sizes for specific components. Thus, the quantitative results should be considered as preliminary outcomes pointing to the most promising components of rehabilitation for vulnerable patients. In addition, the results may play an important role in planning future larger-size hypothesis testing trials within the field of cardiac rehabilitation.

The project is set within a specific Danish context. Collaboration between a hospital, a rehabilitation center, two municipalities, and civil organizations is at the heart of the project. However, culture, procedures, and collaboration vary between settings, and the results from this study need to be locally adjusted when implemented in another setting.

### Sources of potential bias

There is a risk of bias during the recruitment process as patients with vulnerability may be less likely to agree to participate in the study, and when participating may be less likely to fill out

questionnaires at baseline and follow-ups. To diminish this bias, the clinical coordinator is trained in recruiting patients with vulnerabilities. During recruitment, basic characteristics of patients declining to participate in the study are noted. However, we will not be able to assess vulnerability directly, as this is measured in the baseline questionnaire.

It is not possible to blind health professionals and patients regarding the allocation to the two study arms. As a result, patients in the control group may value usual care more negatively, as they perceive the intervention group receives more offers. Project nurses are allocated to manage patients from either the control or intervention group to diminish the risk of cross-pollution between the groups. However, patients may meet patients from other groups as usual care is accessible for all patients. In the analysis of the qualitative data, this is included as a potential influence on the results.

## Benefits

The establishment of a cross-sectoral partnership enabling close collaboration between health-care professionals from different organizations support a more agile and improved sector transition for cardiac patients during a rehabilitation course. In addition, the project used co-creation in the development of the multi-component rehabilitation intervention involving both the patient advisory board and health care professionals from the partnering organizations. This was done in accordance to the literature [52, 53] showing that solutions created through interdisciplinary and cross-sectoral collaboration are more likely to be used by the intended, to reflect the reality of the patient and to be implemented and have an effect in the long term.

The inclusion of the patient advisory board in the development phase is unique, as patients with cardiac diseases due to their rehabilitation experiences can highlight the deficiencies in the current rehabilitation program from a patient perspective.

The useful results of the pilot study are the ability to answer questions about recruitment procedures, sample size estimates required for future studies, how to improve the quality and efficiency of the intervention, the implementation process, and cross-sectoral collaboration which will point out meaningful outcome measures for future trials. To ensure successful implementation and sustainability, adjustments during the implementation will be made and registered.

The qualitative results provide knowledge useful to develop and implement differentiated initiatives in other rehabilitation settings as they provide data about patients' needs and how to match these needs with patient-targeted rehabilitation offers. Also, the quantitative results can point out which efforts in the intervention are the most widely used, although not conclusive, and therefore have the highest potential in future studies.

As we calculate with 100+ patients in each group, we follow the recommendation about sample sizes for external pilot studies which should be at least 30+ per group [54].

## Dissemination

The partnership between the involved organizations will also be utilized during the analysis and dissemination of results and ensure appropriate reflexivity. Members of the research group in CCRP will discuss the findings on quarterly meetings with partners. In addition, the research group will consult the expert group consisting of experienced researchers from clinical departments and universities to support analysis and dissemination.

The dissemination strategy of the study will focus on the communication and implementation of research findings into practice. Our results will be published in peer-reviewed scientific journals and presented at international and national conferences and seminars. Short reports

aimed at practitioners and policymakers in Denmark will be made to provide communication of key findings and implications for practice for relevant stakeholders.

## Conclusion

Results from HeRTA will point to a model that is feasible and sustainable within the Danish healthcare system and forms a coherent rehabilitation pathway for people with cardiac disease. The model can be adjusted locally to fit the context in other rehabilitation locations and to ensure local stakeholder engagement. Our preliminary economic evaluation of financial cost and use of resources will provide a qualification of the sustainability of the model. Chronic patients' concerns and difficulties in terms of rehabilitation are mostly comparable across chronic illnesses. The results can be used as an inspiration within rehabilitation for other chronic illnesses e.g. diabetes, Chronic obstructive pulmonary disease (COPD), and multimorbidity conditions.

HeRTA can form the basis for further targeted rehabilitation studies: a) rigorous RCT studies dissecting the effect of specific elements of the intervention that shows the greatest potential for positive benefits, and b) studies that test the generalizability across contexts to ensure transferability of results.

## Supporting information

**S1 Checklist. SPIRIT 2013 checklist.**
(DOC)

**S1 Protocol.**
(DOCX)

## Acknowledgments

We appreciate Rikke Kart Jacobsen, statistician in the StatPharm Section, CCRP, Frederiksberg Hospital, for helping with the power calculation and the statistical plan for analyses.

## Author Contributions

**Conceptualization:** Hanne Birke, Ida Foxvig, Karin Burns, Ulla Toft, Anders Blædel Gottlieb Hansen, Pernille Ibsen Hauge, Sussie Foghmar, Rikke Bülow Mindegaard, Louise Meinertz Jakobsen.

**Data curation:** Hanne Birke, Karin Burns, Louise Meinertz Jakobsen.

**Formal analysis:** Hanne Birke, Karin Burns, Louise Meinertz Jakobsen.

**Funding acquisition:** Hanne Birke, Ida Foxvig, Louise Meinertz Jakobsen.

**Investigation:** Hanne Birke, Ida Foxvig, Karin Burns, Louise Meinertz Jakobsen.

**Methodology:** Hanne Birke, Ida Foxvig, Karin Burns, Anders Blædel Gottlieb Hansen, Louise Meinertz Jakobsen.

**Project administration:** Hanne Birke, Louise Meinertz Jakobsen.

**Resources:** Hanne Birke, Pernille Ibsen Hauge, Sussie Foghmar, Rikke Bülow Mindegaard.

**Software:** Hanne Birke, Karin Burns.

**Supervision:** Hanne Birke, Karin Burns, Ulla Toft, Anders Blædel Gottlieb Hansen, Louise Meinertz Jakobsen.

**Visualization:** Ida Foxvig, Rikke Bülow Mindegaard, Louise Meinertz Jakobsen.

**Writing – original draft:** Hanne Birke, Ida Foxvig, Karin Burns, Louise Meinertz Jakobsen.

**Writing – review & editing:** Hanne Birke, Ida Foxvig, Karin Burns, Ulla Toft, Anders Blædel Gottlieb Hansen, Pernille Ibsen Hauge, Sussie Foghmar, Rikke Bülow Mindegaard, Louise Meinertz Jakobsen.

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
