## [Decision Letter · Decision Letter 0]

14 Apr 2022

PONE-D-22-05894Heart Rehabilitation for All (HeRTA): Protocol for a feasibility study and pilot randomized trial.PLOS ONE

Dear Dr. Birke,

Thank you for submitting your manuscript to PLOS ONE. After careful consideration, we feel that it has merit but does not fully meet PLOS ONE’s publication criteria as it currently stands. Therefore, we invite you to submit a revised version of the manuscript that addresses the points raised during the review process.

We look forward to receiving your revised manuscript.

Kind regards,

Walid Kamal Abdelbasset, Ph.D.

Academic Editor

PLOS ONE

Journal Requirements:

"The work is funded by The Danish Heart Association, Helsefonden, and Brødrene Hartmann Foundation. The two last finding sources had no role in the design of this study and will not have any role during execution, analyses, interpretation of data, or decision to submit results. The Danish Heart Association is also a partner in the project as they deliver one of the components in the complex intervention. They have responsibility over their own component and have participated in the development on an equal footing with the other partners. The Danish Heart Association will therefore have an important role during execution, but not the interpretation of data, analyses, or decision to submit the results."

We note that you have provided funding informationt. However, funding information should not appear in the Funding section or other areas of your manuscript. We will only publish funding information present in the Funding Statement section of the online submission form. 

"This work was supported by The Danish Heart Association (19-R136-A9125-22129 to HB) https://www.hjerteforeningen.dk , by Brødrene Hartmann (A36031 to HB) https://www.hartmannfonden.dk/, and by Helsefonden (21-B-0016 to HB) https://www.helsefonden.dk. "

Reviewers' comments:

Reviewer's Responses to Questions

**Comments to the Author**

1. Does the manuscript provide a valid rationale for the proposed study, with clearly identified and justified research questions?

Reviewer #1: Yes

Reviewer #2: Yes

2. Is the protocol technically sound and planned in a manner that will lead to a meaningful outcome and allow testing the stated hypotheses?

Reviewer #1: Yes

Reviewer #2: Yes

3. Is the methodology feasible and described in sufficient detail to allow the work to be replicable?

Reviewer #1: Yes

Reviewer #2: Yes

4. Have the authors described where all data underlying the findings will be made available when the study is complete?

Reviewer #1: Yes

Reviewer #2: Yes

5. Is the manuscript presented in an intelligible fashion and written in standard English?

Reviewer #1: Yes

Reviewer #2: Yes

6. Review Comments to the Author

You may also provide optional suggestions and comments to authors that they might find helpful in planning their study.

Reviewer #1: thanks a lot for your manuscript that provide a valid rationale for the proposed study --

Reviewer #2: Well described feasibility study.

There are few things need to be considered:

- Some abbreviation are mentioned in the manuscript without previous introduction such as NGO.

- Why in the inclusion criteria, patient post CABG are not included?

- The model created which include follow up by phone calls, from my point of view is a little bit outdated, since nowadays we are experiencing advancement in Telerehabilitation or remote monitoring of the patients. Has this been considered by the authors?

- Most of the outcome measures are relatively subjective and does not reflect the physiological benefits of attendance and adherence to the program.

- There is a need to identify the phase of cardiac rehabilitation provided at each stage of the model suggested.

7. PLOS authors have the option to publish the peer review history of their article (what does this mean?). If published, this will include your full peer review and any attached files.

Reviewer #1: **Yes: **ahmed abdelmoniem ibrahim

Reviewer #2: No

---

## [Author Response · Author response to Decision Letter 0]

24 May 2022

Response to Reviewers

Journal Requirements:

Comment 1:

Our response 

Throughout the manuscript, we have made corrections according to PLOS ONE’s style requirements:

• All headings: Level 1 heading bold 18 pt font, Level 2 Heading bold 16 pt font, Level 3 heading bold 14 pt font

• Figure has been corrected to Fig

• Fig titles has been corrected to bold types

• The figure files has been renamed fig1.tiff etc.

• References has been corrected to PLOS ONE reference style

Comment 2:

Thank you for stating the following in the Funding Section of your manuscript: 

"The work is funded by The Danish Heart Association, Helsefonden, and Brødrene Hartmann Foundation. The two last finding sources had no role in the design of this study and will not have any role during execution, analyses, interpretation of data, or decision to submit results. The Danish Heart Association is also a partner in the project as they deliver one of the components in the complex intervention. They have responsibility over their own component and have participated in the development on an equal footing with the other partners. The Danish Heart Association will therefore have an important role during execution, but not the interpretation of data, analyses, or decision to submit the results."

We note that you have provided funding information. However, funding information should not appear in the Funding section or other areas of your manuscript. We will only publish funding information present in the Funding Statement section of the online submission form. 

"This work was supported by The Danish Heart Association (19-R136-A9125-22129 to HB) https://www.hjerteforeningen.dk , by Brødrene Hartmann (A36031 to HB) https://www.hartmannfonden.dk/, and by Helsefonden (21-B-0016 to HB) https://www.helsefonden.dk. "

Our response 

We have accordingly removed funding information from the manuscript. We wish the following funding statement, which is also described in our cover letter, in the online submission form:

"This work was supported by The Danish Heart Association (19-R136-A9125-22129 to HB) https://www.hjerteforeningen.dk , by Brødrene Hartmann (A36031 to HB) https://www.hartmannfonden.dk/, and by Helsefonden (21-B-0016 to HB) https://www.helsefonden.dk. The latter funding sources had no role in the design of this study and will not have any role during execution, analyses, interpretation of data, or decision to submit results. The Danish Heart Association is also a partner in the project as they deliver one of the components in the complex intervention. They have responsibility for their own component and have participated in the development on an equal footing with the other partners. The Danish Heart Association will therefore have an important role during execution, but not the interpretation of data, analyses, or decision to submit the results."

Comment 3:

We note that you have stated that you will provide repository information for your data at acceptance. Should your manuscript be accepted for publication, we will hold it until you provide the relevant accession numbers or DOIs necessary to access your data. If you wish to make changes to your Data Availability statement, please describe these changes in your cover letter and we will update your Data Availability statement to reflect the information you provide.

Our response 

We wish to make the following change to our Data Availability Statement, which is also described in our cover letter, as we have become aware of restrictions related to data availability according to Danish registrations:

“The participants in this study have not given consent to make these data publicly available. Therefore, the data and the information regarding the participants cannot be publicly available. A request for access to the data needs approval from appropriate Danish authorities and is subject to Danish regulations on personal data protection. A request for an arrangement of data transfer agreements can be sent to the corresponding author Hanne Birke or to author Louise Meinertz Jakobsen.”

Comment 4:

Your ethics statement should only appear in the Methods section of your manuscript. If your ethics statement is written in any section besides the Methods, please move it to the Methods section and delete it from any other section. Please ensure that your ethics statement is included in your manuscript, as the ethics statement entered into the online submission form will not be published alongside your manuscript.

Our response 

The ethics section has been removed from page 23 to page 10 in the Methods section.

Comment 5:

Our response 

We have not used track changes in the following reference corrections in the reference list 

as it can create problems with the reference manager system Mendeley. The corrections are:

Reference 1 was by mistake not written correctly in the introduction– it is now corrected from 

“(ref who)” to “1” on page 6, line 91. 

Reference nr 3: Broge C. Op imod en halv million hjertepatienter i 2020. Copenhagen, has been

retracted from the website and therefore been exchanged to 

“The Danish Heart Association. Facts about cardiovascular disease in Denmark (in Danish). [cited22 

Apr 2022]. Available: https://hjerteforeningen.dk/alt-om-dit-hjerte/noegletal/”

Reference 11: “The Capital Region of Copenhagen. Cardiac rehabilitering (in Danish), (2019).” has 

been corrected to 

“The Capital Region of Copenhagen. Cardiac rehabilitering (in Danish). 2019. 

Available: https://www.regionh.dk/til-fagfolk/Sundhed/Tvaersektorielt-samarbejde/kronisk-

sygdom/Forløbsprogrammer/Documents/2019 Hjerteforloebsprogram.pdf”

Reference 15: “Frederiksen HW. Cardiac Rehabilitation Among Migrants. A Mixed-Methods Study. 

phd thesis 2018; 87.” has been corrected to an article from the phd thesis

“ Al-Sharifi F, Winther Frederiksen H, Knold Rossau H, Norredam M, Zwisler AD. Access to cardiac 

rehabilitation and the role of language barriers in the provision of cardiac rehabilitation to 

migrants. BMC Health Serv Res. 2019;19: 1–8. doi:10.1186/s12913-019-4041-1.”

Reference 22: “The World Health Organization (WHO). Declaration of Alma-Ata, 

https://www.who.int/publications/almaata_declaration_en.pdf (1978, accessed 29 January 

2021).” has been changed in the website and now corrected to 

“The World Health Organization. Declaration of Alma-Ata. 1978 [cited 29 Jan 2021]. Available: 

https://www.who.int/teams/social-determinants-of-health/declaration-of-alma-ata”

Reference 30: “Vichealth. The partnerships analysis tool. Vichealth.vic.gov.au 2016; 8.” has been 

corrected to 

“ Vichealth. The partnerships analysis tool. In: Vichealth.vic.gov.au [Internet]. 2016 

[cited 2 Feb 2021] p. 8. Available: https://www.vichealth.vic.gov.au/search/the-partnerships-

analysis-tool.”

Reference 31: “. Forebyggelseslaboratorium DS. Et værktøj til partnerskabsanalyse. En ressource til 

etablering, udvikling og vedligeholdelse af partnerskaber indenfor sundhedsfremme., 

https://www.regionh.dk/forebyggelseslaboratoriet/nyheder/Documents/værktøj til 

partnerskabsanalyse.pdf (2020, accessed 11 February 2021). has been corrected to

“Det sektorfri Forebyggelseslaboratorium. Et værktøj til partnerskabsanalyse. En ressource til 

etablering, udvikling og vedligeholdelse af partnerskaber indenfor sundhedsfremme (in Danish). 

2020 [cited 11 Feb 2021]. Available: 

https://www.regionh.dk/forebyggelseslaboratoriet/nyheder/Documents/værktøj til 

partnerskabsanalyse.pdf”

Reference 42: “Malterud K. Kvalitative metoder i medicinsk forskning: en innføring (3. edition). In 

Norwegian. Oslo: Universitetsforlaget, 2011.” Has been corrected to

 “ Malterud K. Kvalitative metoder i medicinsk forskning: en innføring (3. edition) (in Norwegian). 

Oslo: Universitetsforlaget; 2011.”

Reference 50: “clinical trials, https://clinicaltrials.gov/ (2021, accessed 1 December 2021)” has 

been corrected to

“ClinicalTrials.gov. [cited 1 Dec 2021]. Available: https://clinicaltrials.gov/”

Reviewer #1: thanks a lot for your manuscript that provide a valid rationale for the proposed study --

Reviewer #2: 

Comment 6:

- Some abbreviation are mentioned in the manuscript without previous introduction such as NGO.

Our response 

On page 9, line 142-143, MRC has been corrected to “UK Medical Research Council (MRC)”

On page 9, line 144-145 NGO has been corrected to “a non-governmental organization”

On page 10, lines 182-183, we have added “randomized controlled trial (RCT)-component” 

On page 13, lines 249-250, we have added “The Victorian Health Promotion Foundation

 (VicHealth)”

On page 20, line 404, we have added “the 12-Item Short Form Survey (SF-12)”

On page 28, lines 594-595, we have added “Chronic obstructive pulmonary disease (COPD)”

Comment 7 

- Why in the inclusion criteria, patient post CABG are not included?

Our response

Post coronary artery bypass graft (CABG) patients are included in the group diagnosed with ischemic heart disease just like patients who have undergone PCI (percutaneous coronary intervention).

Comment 8

- The model created which include follow up by phone calls, from my point of view is a little bit outdated, since nowadays we are experiencing advancement in Telerehabilitation or remote monitoring of the patients. Has this been considered by the authors?

Our response

Telerehabilitation or remote monitoring of patients is currently not an option in the Cardiac Outpatient Clinic at Hvidovre Hospital and at the Rehabilitation Center Albertslund. Using telerehabilitation requires introduction and guidance to the patients to enable this practice. In general, telerehabilitation is used in the treatment and monitoring of diseases. We consider phone calls an appropriate, easily accessible, and low-cost method to follow up on the patients’ physical activity level in this study which primarily focuses on participation in a cross-sectoral rehabilitation program and long-term physical activity.

Comment 9 

- Most of the outcome measures are relatively subjective and does not reflect the physiological benefits of attendance and adherence to the program.

Our response

We agree that increased physical activity level among the participants can have a physiological effect and that it, therefore, could be relevant to add objective measures reflecting the physiological benefits. However, the interventions in this study focus on increasing rehabilitation participation rates and long-term physical activity levels rather than improving physiological benefits by increasing the amount of exercise or intensity of the activity. Therefore, we have chosen not to measure objective biological markers of physiological effects of participation in rehabilitation. Although most of our outcome measures are self-reported, we also measure participation through registration by partners. 

Comment 10

- There is a need to identify the phase of cardiac rehabilitation provided at each stage of the model suggested.

Our response

According to the Danish Cardiac Rehabilitation Program, cardiac rehabilitation includes three phases:

• Phase one is the acute phase with hospital admission, clarification of diagnosis, and treatment (medical and/or surgical). The duration is days.

• Phase two is the period after discharge from the hospital, patients are offered further treatment and initiation of a rehabilitation program for either 12 weeks in the hospital, 12 weeks in the municipality, or 6 weeks in the hospital and 6 weeks in the municipality. The duration is weeks to months.

• Phase three is a further follow-up, control, and continued medical treatment by a general practitioner and rehabilitation efforts in the municipality. The duration is life-long.

Based on these phases, the interventions in this study is placed in phases two and three. We have, however, chosen not to define the different phases and stages of cardiac rehabilitation in our rehabilitation model, as we consider our rehabilitation model as one coherent course with individual options. Our participants can select or deselect some of the rehabilitation offers during the program. They can choose, for example, to go directly from the hospital to a sports association in civil society. Since all participants do not have to follow the same rehabilitation path, we believe that a division into phases is not appropriate in the current study. 

In addition

We have corrected the definition of participation in rehabilitation in the abstract on page 5, lines 78-80 from 

“The primary outcome: rehabilitation participation (attending ≥ two activities: pro-active counselling, patient education, smoking cessation, dietary counseling, physical exercise, local sports association activity) and reaching at least 50% attendance.” 

to:

“The primary outcome: rehabilitation participation (attending ≥ two activities: patient education, smoking cessation, dietary counseling, and physical training) and reaching at least 50% attendance.”

The reason for the correction is that pro-active counselling and local sports association activity by mistake was written in the definition.

---

## [Editor Report · Decision Letter 1]

6 Jun 2022

Heart Rehabilitation for All (HeRTA): Protocol for a feasibility study and pilot randomized trial.

PONE-D-22-05894R1

Dear Dr. Birke,

We’re pleased to inform you that your manuscript has been judged scientifically suitable for publication and will be formally accepted for publication once it meets all outstanding technical requirements.

Kind regards,

Walid Kamal Abdelbasset, Ph.D.

Academic Editor

PLOS ONE

Additional Editor Comments (optional):

All comments have been addressed by the authors. No further comments are required.
---

## [Editor Report · Acceptance letter]

8 Jun 2022

PONE-D-22-05894R1 

Heart Rehabilitation for All (HeRTA): Protocol for a feasibility study and pilot randomized trial. 

Dear Dr. Birke:

I'm pleased to inform you that your manuscript has been deemed suitable for publication in PLOS ONE. Congratulations! Your manuscript is now with our production department. 

Kind regards, 

on behalf of

Dr. Walid Kamal Abdelbasset 

Academic Editor

PLOS ONE